# Adversarially Trained Neural Policies in the Fourier Domain

**Ezgi Korkmaz** [1]

## Abstract

Reinforcement learning policies based on deep neural networks are vulnerable to imperceptible adversarial perturbations to their inputs, in much the same way as neural network image classifiers. Recent work has proposed several methods for adversarial training for deep reinforcement learning agents to improve robustness to adversarial perturbations. In this paper, we study the effects of adversarial training on the neural policy learned by the agent. In particular, we compare the Fourier spectrum of minimal perturbations computed for both adversarially trained and vanilla trained neural policies. Via experiments in the OpenAI Atari environments we show that minimal perturbations computed for adversarially trained policies are more focused on lower frequencies in the Fourier domain, indicating a higher sensitivity of these policies to low frequency perturbations. We believe our results can be an initial step towards understanding the relationship between adversarial training and different notions of robustness for neural policies.

## 1. Introduction

Deep neural networks (DNNs) have been initially employed in deep reinforcement learning by Mnih et al. (2015) to approximate the state-action value function for large action size or state size MDPs. With this initial success deep reinforcement learning became an emerging subfield with many applications including autonomous driving, robotics Kalashnikov et al. (2018), financial trading Noonan (2017), biomedical Daochang & Jiang (2018); Huan-Hsin et al. (2017), pharmaceuticals and drug design Gogineni et al. (2020); Xu et al. (2020).

While the successes of DNNs grew, a line of research focused on their reliability and robustness. Initially, Szegedy et al. (2014) demonstrated that it is possible to fool image classifiers by adding visually imperceptible perturbations to

neural network inputs. Follow up work by Goodfellow et al. (2015) showed that these perturbations demonstrate that neural networks are learning approximately linear functions. Several studies focused on overcoming this susceptibility towards specifically crafted visually imperceptible perturbations, and proposed several strategies focusing on training neural networks against these particular malicious perturbations Madry et al. (2018). More recent study showed that much of the work focusing on adversarial training suffers from the obfuscated gradient problem Athalye et al. (2018). Further, while there is a significant amount of study focusing on adversarial training, several other works suggest that adversarial perturbations may be inevitable Dohmatob (2019); Mahloujifar et al. (2019); Gourdeau et al. (2019).

For these reasons in this paper we focus on adversarially trained neural policies and make the following contributions:

- We investigate the frequency domain of the minimal perturbations produced by the Carlini & Wagner (2017) formulation for both state-of-the-art adversarially trained models and vanilla trained models.

- We conduct multiple experiments in the OpenAI Atari Baselines.

- We show that the perturbations produced from state-of-the-art adversarially trained models are suppressed in high frequencies and more concentrated in lower frequencies in the Fourier domain compared vanilla trained neural policies.

For additional results and completeness see the full version of this paper published in Korkmaz (2021d). See Korkmaz (2021a;b;c; 2023) for more issues introduced by adversarial training in deep reinforcement learning.

## 2. Background

### 2.1. Deep Reinforcement Learning

The reinforcement learning problem is described as Markov Decision Process (MDP) consisting of the tuple $(S, A, r, \mathcal{P}, \gamma)$ where $S$ is the set of states, $A$ is the set of actions, $r$ represents the reward function, $\mathcal{P}$ represents the state transition probabilities and $\gamma$ represents the discount

---

[1] Ezgi Korkmaz <ezgikorkmazmail@gmail.com>

*Proceedings of the 37th International Conference on Machine Learning*, PMLR 108, 2021. Copyright 2021 by the author(s).

factor. In reinforcement learning the goal is to learn a policy $\pi(s, a)$ that maximizes the expected cumulative rewards of the agent. $Q$-learning achieves this goal via the iterative Bellman update and builds a state action value function for each state action pair. Note that in our paper environments with high dimensional state spaces are considered and evaluated.

## 2.2. Methods for Producing Adversarial Examples

Manipulating the output of neural networks by introducing imperceptible perturbations has been introduced by Szegedy et al. (2014) based on a box constrained optimization method.

$$\underset{x_{\text{adv}}}{\arg\min} = c \cdot \|x_{\text{adv}} - x\| - J(x_{\text{adv}}, y) \qquad (1)$$

While this proposed method is computationally expensive, Goodfellow et al. (2015) proposed a faster and simpler method based gradients in a nearby $\epsilon$-ball,

$$x_{\text{adv}} = x + \epsilon \cdot \frac{\nabla_x J(x, y)}{\|\nabla_x J(x, y)\|_p}, \qquad (2)$$

where $y$ represents the labels, $x$ represents the input, and $J(x, y)$ represents the cost function used to train the network. Kurakin et al. (2016) propose to search iteratively inside this $\epsilon$-ball with the fast gradient sign method (FGSM) proposed by Goodfellow et al. (2015).

$$x_{\text{adv}}^0 = x, \qquad (3)$$
$$x_{\text{adv}}^{N+1} = \text{clip}_\epsilon(x_{\text{adv}}^N + \alpha \text{sign}(\nabla_x J(x_{\text{adv}}^N, y))) \qquad (4)$$

Madry et al. (2018) explained adversarial training in terms of the theory of robust optimization, and referred to this class of iterative methods used to produce adversarial perturbations as projected gradient descent (PGD). While several other adversarial optimization methods are proposed based on momentum based iterative updates Korkmaz (2020), Carlini & Wagner (2017) formulated this problem in a more targeted way, and proposed a method based on distance minimization for a given label in image classification. For deep reinforcement learning this formulation is based on distance minimization for a given a target action which is not equal to the best action decided by the trained policy,

$$\min_{s_{\text{adv}} \in D_{\epsilon, p}(s)} \quad \|s_{\text{adv}} - s\|_p$$
$$\text{subject to} \quad a^*(s) \neq a^*(s_{\text{adv}}),$$

Note that $a^*(s)$ denotes the action taken in state $s$, and $a^*(s_{\text{adv}})$ denotes the action taken under the influence of

adversarial perturbation (i.e. in state $s_{\text{adv}}$). Athalye et al. (2018) showed that the Carlini & Wagner (2017) adversarial formulation can break several proposed defenses. For this reason, in this paper we will focus on perturbations produced by the Carlini & Wagner (2017) formulation.

## 2.3. Perturbation Formulations and Adversarial Training

The first studies on adversarial examples for deep reinforcement learning were Huang et al. (2017) and Kos & Song (2017). Both papers focused on using FGSM perturbations applied to the state observations in order to degrade the performance of trained neural policies. In the other direction, Mandlekar et al. (2017) introduced a form of adversarial training for deep neural policies by modifying the inputs at training time with perturbations based on the gradient of the cost function. Since reinforcement learning involves interaction with the environment there has also been effort to incorporate this interaction into adversarial training. Both Pinto et al. (2017) and Gleave et al. (2020) use game-theoretic models of the interaction between the deep reinforcement learning agent and then design training algorithms based on playing against the adversary in a zero-sum game in order to improve robustness. Recent work by Zhang et al. (2020) involves modifying the original MDP to create what they term a state-adversarial MDP. The authors then design theoretically-motivated adversarial training algorithms for deep neural policies based on training within the state-adversarial MDP. Recently, Korkmaz (2022a) showed that the deep reinforcement learning policies learn shared non-robust features intrinsic to the learning environment across MDPs. Quite recently, Korkmaz (2023) demonstrated that natural directions that are intrinsic to the MDP can cause similar impact on the policy performance with higher perceptual similarity than adversarial directions. While this study questions the definition of robustness in the certified adversarial training algorithms, Korkmaz (2023) further demonstrates that the state-of-the-art adversarial training techniques blocks the generalization capabilities of deep reinforcement learning policies[1]. Further see here for a detailed categorization and the unification of generalization in deep reinforcement learning (Korkmaz, 2024).

## 3. Experimental Setup

In our experiments we use OpenAI Brockman et al. (2016) Atari baselines introduced by Bellemare et al. (2013). The deep neural policies are trained with DDQN Wang et al. (2016) and State-Adversarial DDQN (SA-DDQN) Zhang et al. (2020). We test trained policies averaged over 10

---

[1]There is also some recent work focusing on the robustness problems in deep inverse reinforcement learning. See these studies for more details (Korkmaz, 2022b;c).

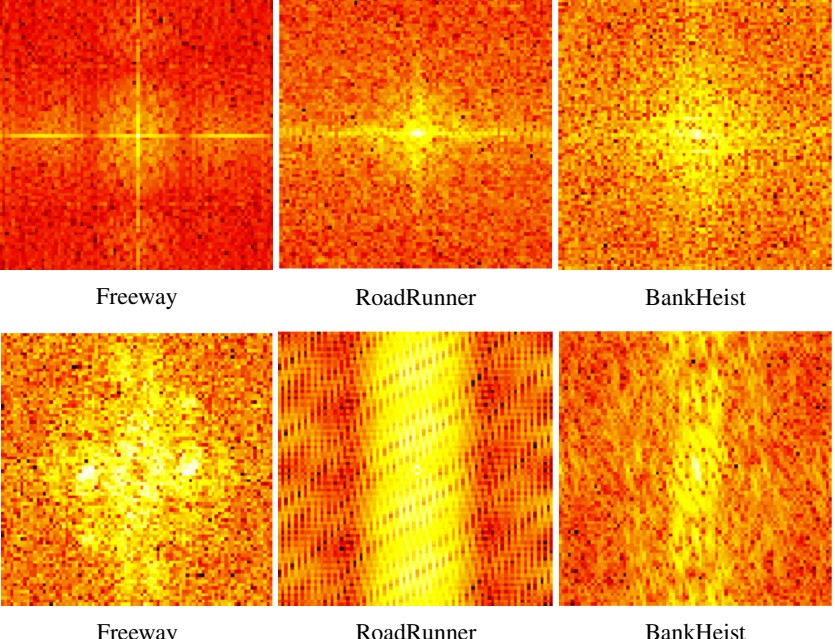

Freeway           RoadRunner           BankHeist

Freeway           RoadRunner           BankHeist

*Figure 1.* Fourier spectrum of the perturbations computed via the Carlini & Wagner (2017) formulation for state-of-the-art adversarially trained models and vanilla trained models. First Row: Adversarially trained. Second Row: Vanilla trained.

episodes. Note that SA-DDQN is certified against $\ell_\infty$-norm bounded perturbations at $\epsilon = 1/255$. Therefore, we also bound the computed perturbations by this threshold and find the perturbations with $\ell_\infty$-norm lower than this value.

## 4. Neural Policy Perturbations in Fourier Domain

In this paper we conduct an investigation on the frequency domain of the perturbations computed from vanilla trained agents and adversarially trained agents. In particular we conduct the following experiment: For a given state compute a minimum length perturbation which causes the agent to change its optimal learned action. If the minimal perturbation has norm smaller than a given threshold, compute the Fourier transform of the perturbation and record this data. By comparing the results of this experiment for adversarially trained versus vanilla trained agents, we can understand the affects of adversarial training on the directions to which the neural policy is sensitive. We now describe the details of the experimental setup.

For the adversarially trained agents, we focus on the state-of-the-art adversarial training algorithm SA-DQN proposed by Zhang et al. (2020). In this study the authors model the interaction between the neural policy and the introduced perturbations as a state-adversarial modified Markov Decision Process (MDP). The authors claim that the agents trained in the SA-MDP with the proposed algorithm SA-DQN are

more robust to adversarial perturbations and natural noise introduced to the agent's perception system. Furthermore, the authors demonstrate the robustness of SA-DQN against perturbations produced by the PGD attack proposed by Madry et al. (2018).

To compute the minimum length perturbations for our experiments we use the Carlini & Wagner (2017) formulation, which searches for a perturbation of minimum length that causes the agent to change its optimal action. For each state we use the Carlini & Wagner (2017) method to produce a perturbation $\eta = s_{\text{adv}} - s$. Note that the certified bound for Zhang et al. (2020) is $\frac{1}{255}$; therefore, we ensure the perturbation produced has norm $\|\eta\|_\infty < \frac{1}{255}$. We then compute the Fourier transform of $\eta$ and add it to our dataset. Note that it is possible to break the certified defense via Carlini & Wagner (2017) perturbations. The "certified defense" proposed by Zhang et al. (2020) only holds for a fraction of states.

In Figure 1 we visualize the Fourier transform of a minimal perturbation for both vanilla trained and adversarially trained agents in RoadRunner, BankHeist and Freeway. The center of each image corresponds to the Fourier basis function where both spatial frequencies are zero, and the magnitude of the spatial frequencies increases as one moves outward from the center. From these visualizations it is clear that the perturbations for the adversarially trained models have their Fourier transform concentrated at lower frequencies than those of the vanilla trained models. To

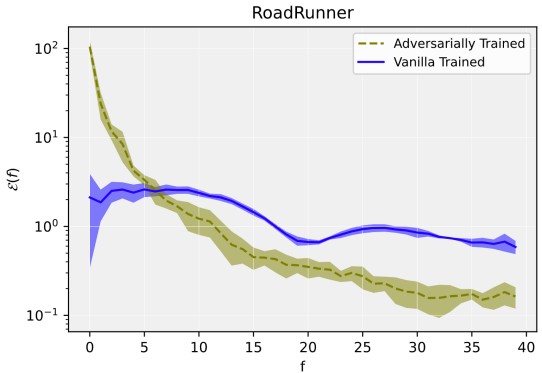 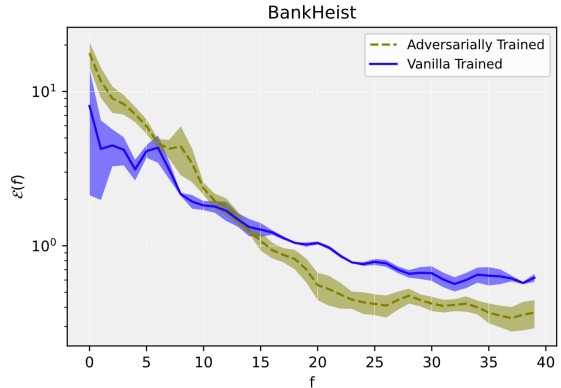

*Figure 2.* Perturbations computed via the Carlini & Wagner (2017) formulation for state-of-the-art adversarially trained deep neural policies and vanilla trained deep neural policies in Fourier domain for RoadRunner and BankHeist environments.

make this claim formal, for each number $f$ we compute the total energy $\mathcal{E}(f)$ of the Fourier transform for basis functions whose maximum spatial frequency is equal to $f$. In Figure 2 and Figure 3 we plot the average of $\mathcal{E}(f)$ over the minimal perturbations computed in our experiments.

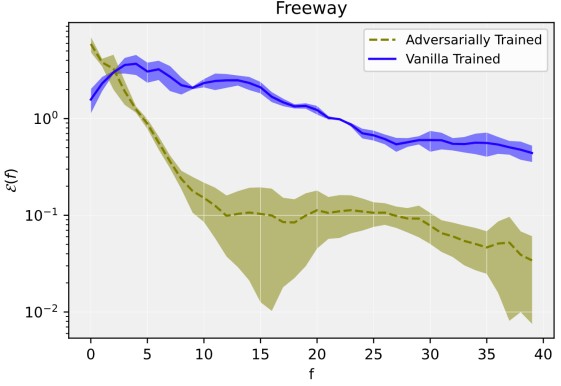

*Figure 3.* Perturbations computed via the Carlini & Wagner (2017) formulation for state-of-the-art adversarially trained models and vanilla trained models in Fourier domain for Freeway environment.

We find that the minimal perturbations computed for adversarially trained neural policies are indeed shifted towards lower frequencies when compared to those for vanilla trained neural policies. This shift in the frequency domain of the computed perturbations implies that adversarially trained neural policies may be more sensitive towards low frequency perturbations. Hence, naturally this brings out questions of robustness claims on adversarial training. While adversarially trained agents do gain robustness against some portion of the spectrum (i.e. higher frequencies) the adversarially trained deep neural policies become more vulnerable to different portion of the spectrum (i.e. lower frequencies).

## 5. Conclusion

In this paper we focused on perturbations computed from state-of-the-art adversarially trained neural policies. We conducted several experiments and compared adversarially trained models to vanilla trained models. We investigated the frequency domain of the perturbations computed from state-of-the-art adversarially trained neural policies and vanilla trained neural policies. We found that the perturbations computed from adversarially trained models are more concentrated in lower frequencies compared to the vanilla trained neural policies. We believe this initial work outlines the vulnerabilities of adversarially trained neural policies and can be initial step towards building robust and reliable deep reinforcement learning agents.

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
