# OpenReview forum: "Adversarially Trained Neural Policies in the Fourier Domain"
_ICML.cc/2021/Workshop/AML — ICML 2021 Workshop AML Poster_

### Official Review · Reviewer_DrcP · 2021-06-20
**The results are interesting but one-sided**

**Rating:** Accept
**Confidence:** 5

**Review:**

This paper compare the perturbations generated in adversarial training agent and vanilla DRL agent in different frequencies in the Fourier domain. The empirical results shows that the adversarial-training is more vulnerable to low-frequency perturbation in Fourier domain provided by targeted attack.

Main Concerns:
1. need more victim policies and more adversarial-trained policies. I also suggest to test on MuJoCo environments.
2. other kinds of perturbations.  My main concern of this work is whether this conclusions holds for other kinds of perturbations.

---

### Decision · Program_Chairs · 2021-06-21

**Decision:**

Accept (Poster)

**Comment:**

This paper studied adversarial attacks on deep RL policies. There are some concerns raised by the reviewers, which could be addressed in the revision.